# Patient organisations' views, motivations and experiences on patient involvement in cancer research: a pilot study in Portugal

Constança Roquette ,[1] Sofia Crisóstomo ,[2,3] Tamara Milagre,[4] Rute Simões Ribeiro,[5] Ana Rita Pedro ,[6] André Valente[7]

¹Nova School of Business and Economics, Universidade NOVA de Lisboa, Carcavelos, Portugal
²ISCTE - Instituto Universitário de Lisboa, Lisboa, Portugal
³GAT - Grupo de Ativistas em Tratamentos, Lisbon, Portugal
⁴EVITA - Cancro Hereditário, Lisbon, Portugal
⁵NOVA University Lisbon NOVA National School of Public Health, Lisbon, Portugal
⁶Public Health Research Centre, Comprehensive Health Research Center (CHRC), NOVA University Lisbon NOVA National School of Public Health, Lisboa, Portugal
⁷Champalimaud Foundation, Lisbon, Portugal

**Correspondence to**
Constança Roquette;
constanca.roquette@novasbe.pt

## ABSTRACT

**Objectives** To characterise Portuguese cancer-related patient organisations and analyse their views, motivations and experiences on patient involvement in cancer research.

**Design** A multistakeholder group, comprising patient representatives and researchers, codesigned a questionnaire after a literature review, online sessions and collaborative mind maps. In May 2021, a survey was conducted among representatives of Portuguese cancer-related patient organisations, focusing on four dimensions: experience in cancer research; perception of its outcomes; motivations and expectations on patient involvement in research; and organisation characteristics.

**Participants** Twenty-seven representatives from Portuguese cancer-related patient organisations responded to the questionnaire, corresponding to a 64% response rate.

**Results** Among the surveyed organisations, 26% have participated in clinical studies. Their involvement occurred in few stages of the research process and, mostly, with low levels of engagement. They showed 'great interest' in participating in most research steps, although this is not reflected in a high perception of influence over these same steps. More than half claimed to have contributed to the increase in patient recruitment and to a better understanding of informed consent by patients involved. Ensuring that research results are more aligned with the true needs of patients is the greatest motivation. Also, our results suggest that the organisation's number of employees and its integration into a European/International network play a relevant role in patient involvement in research.

**Conclusions** This study provides the first in-depth characterisation of Portuguese cancer-related patient organisations and their views, motivations and experiences on patient involvement in cancer research. Most importantly, this study revealed that most of these organisations show great interest in being involved in different R&D stages to ensure that research results are aligned with patients' needs. Their motivation should be turned into greater and more meaningful involvement in practice, so that the cancer community can benefit from the outcomes of truly patient-centred research.

## STRENGTHS AND LIMITATIONS OF THIS STUDY

⇒ A four-domain questionnaire was developed to characterise Portuguese cancer-related patient organisations and their views, motivations and experiences regarding patient involvement in cancer research.

⇒ This research methodically investigates the level of patient involvement in 16 R&D opportunities, going beyond previous studies that mainly focus on identifying the presence or absence of patient involvement through the R&D process.

⇒ This study was a multistakeholder cocreation, having been ideated, designed and conducted with and for patients since the very beginning.

⇒ Although there was a 64% response rate, the universe of cancer-related patient organisations in Portugal is small, and the respondents' number<30, which may compromise the statistical significance of the study, preventing from making broader conclusions.

## INTRODUCTION

In the early 1980s,[1] people with HIV joined voices and advocacy efforts to be involved at all levels of decision-making that affected them, from policy-making to research decisions,[2 3] including identification of gaps and priorities, design, planning and conduct of research itself, as well as decisions on access to experimental medicines and subsequent regulatory authorisation procedures.[4] This call for meaningful involvement in decision-making was institutionalised under the motto 'nothing about us, without us',[3] which was by the same time also adopted by the disability rights movement[5] and some years later appropriated by social movements in other disease areas, including cancer, Alzheimer disease, mental health and many others. In a recent Eurobarometer survey, 'most respondents agreed (61%) that involving nonscientists in research and technological development

ensures that science and technology respond to the needs, values and expectations of society'.[6]

Patient involvement in research and care in general is key to closing the gap between beliefs and perception from researchers, care professionals and other decision-makers and the true needs and preferences of those affected.[7–9] By contributing to optimise research and care, patient involvement leads to more successful responses and consequently improved outcomes that matter and make a difference to people, the so-called people-centred research and care,[10] while strengthening health governance and responsiveness, and improving the quality of decision-making and facilitating its implementation. Patient organisations have been playing a key role in giving voice to patients' needs while building collaborative networks between patient advocates, researchers, clinicians and industry, and steering research and development (R&D) activity, particularly in the context of rare diseases.[11–13]

Even though there has been an increased interest and research on involvement frameworks and benefits, and practical recommendations have been issued by various stakeholders, patient involvement in clinical research is not yet a standard, meaningful and systematic practice[14–23] and several challenges are identified. There is a lack of clarity of the concept of 'patient involvement', which leads to challenges for operationalising and measuring its impacts,[24] and very often involvement is mostly done through consultation, instead of direct engagement, being rarely extended to later stages of the R&D process.[25–27] Matching patient advocates with research opportunities for involvement is also challenging, particularly when the dissemination of these opportunities is limited.[28 29] Finally, there is a lack of or fragmented reporting and critical reflection on patient involvement activities,[27 29] which calls for more research that compares, contrasts and evaluates different patient involvement initiatives.[30]

To address this, on 13 October 2020, Germany, Portugal and Slovenia (the trio Presidency of the Council of the European Union, at the time) signed together the *Europe: Unite against Cancer* declaration to join approaches on strengthening cancer research in Europe and to ensure that patient involvement becomes a standard practice.[31] Moreover, fundamental principles for successful patient involvement in cancer research were largely discussed and determined, under a German Federal Ministry of Education and Research initiative.[32]

Aligned with these initiatives and focused on the Portuguese context as a pilot project, this study aims to understand patient involvement in cancer research landscape and to identify existing good practices, as well as opportunities and challenges for improvement from a bottom-up perspective. Cancer-related patient organisations' involvement at the different stages of medicines R&D is characterised, based on opportunities for involvement described in the roadmap proposed by Geissler *et al*.[14] Organisations' views, and motivations regarding patient involvement in clinical research activities are also analysed. Lastly, descriptive analysis of the Portuguese cancer-related patient organisations is presented. Previous works that analysed Portuguese patient organisations' missions, activities, perspectives and communication strategies on a large scale[33 34] did not present a particular focus on cancer-related patient organisations neither a perspective on these organisation's involvement in research and are probably outdated in face of the increasing interest and awareness on patient involvement among patient organisations and other stakeholders.[35 36]

## METHODS

In December 2020, encouraged by the *Europe: Unite against Cancer* declaration, a multistakeholder collaborative group with six people was put in place with both patient representatives and researchers, aiming to develop and conduct a survey that would provide state-of-the-art information on the opportunities and level of involvement in research of cancer-related patient organisations in Portugal, as well as, to characterise these, identify their views, motivations and experiences. The multistakeholder research team included two patient advocates, one from *EVITA Cancro Hereditário* (a patient organisation focused on hereditary cancer) and another from *Mais Participação, Melhor Saúde* (a community-based action-research collaborative platform), as well as three researchers from *Universidade NOVA de Lisboa* (NOVA School of Business and Economics and the Portuguese National School of Public Health) and one researcher from the Champalimaud Foundation. The survey development timeline is presented in figure 1.

### Questionnaire cocreation

Through the creation of an online collaborative mind map, the work group identified the dimensions to be considered in the survey, as well as the variables and indicators to be included. Four main domains were consensually selected by all group members: (1) experience in cancer research; (2) perception of the outcomes of that experience; (3) motivations and perspectives on patient involvement in cancer research; (4) organisation characteristics.

After several work sessions, the questionnaire structure was agreed, as well as the questions and answer options that would best allow to conduct the intended analysis. The questionnaire final version has 41 questions, an approximate time of completion of 30 min, and is attached as a online supplemental files 1; 2 (questionnaire Portuguese version and English translation files). Most of the questions are closed-ended and multiple-choice questions. For questions about frequency, degree of influence and interest, a four-point Likert scale was used to avoid a neutral option. Open-ended questions are intended not to limit the information collected and give the respondents the opportunity to share their experience.

### Categorisation of opportunities for patient involvement and patient involvement level

In order to understand how Portuguese cancer-related patient organisations have been involved in research,

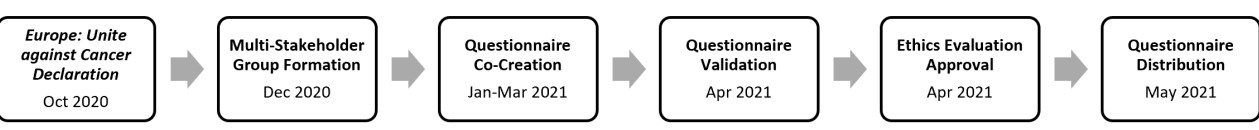

**Figure 1** Survey development timeline.

the roadmap for patient involvement in medicines R&D proposed by Geissler *et al*[14] was used as reference. This roadmap categorises the R&D life cycle into 4 main stages and highlights 16 specific opportunities for patient involvement (table 1), requiring different levels of expertise in the disease area. These opportunities were used to identify those where the surveyed organisations may have had experience with.

Moreover, within each R&D stage and for each specific opportunity, the respondents were asked to rank their highest level of involvement. For that purpose, a scale of involvement was developed (table 2) based on the adaptation of the levels of involvement proposed in the Health

Canada Policy Toolkit for Public Involvement in Decision Making.[37]

### Target sample and participants' eligibility criteria
The study focused on representatives from cancer-related patient organisations in Portugal as the target sample. Identification of these organisations was primarily conducted by the coauthor patient representative from *EVITA Cancro Hereditário*, as there is currently no comprehensive public database of patient organisations in Portugal. The initial list was supplemented through a web search to identify new organisations and verify the status of those believed to be inactive. The final sample

**Table 1** Opportunities for patient involvement in medicines R&D adapted from Geissler *et al*[14] roadmap

| R&D stage | Opportunities for patient involvement in medicines R&D |
|---|---|
| Research priorities | Identification of patients' unmet needs. |
| Research design and planning | Study's synopsis design: design and target population definition. |
| | Protocol design: endpoints, inclusion and exclusion criteria, quality-of-life measures and patient-reported outcomes, crossover, ethical and data protection issues, recruitment and dissemination plan, retention measures, risk–benefit balance, and so on. |
| | Practical considerations: contractual issues, travel expenses and support for family members. |
| | Fundraising for the research project. |
| | Patient information leaflet about the project: content, visual design, readability, language and dissemination. |
| | Informed consent: content, visual design, readability and language. |
| | Ethical review of the research project. |
| Research conduct and operations | Investigators meetings to monitor the project: representation of patients' perspectives on the study, recruitment, challenges, opportunities, which can trigger amendments. |
| | Trial Steering Committee: protocol follow-up, improvements in patient access and adherence to the study. |
| | Data and Safety Monitoring Committee: risk/benefit assessment, drop-out issues and amendments to the project. |
| | Information to participants: communication of protocol amendments and new safety information. |
| | Study reporting: summary of interim results and dissemination in patient community. |
| Dissemination, communication, post-approval | Health Technology Assessment: assessment of value, patient-relevant outcomes and patient priorities. |
| | Regulatory affairs: preparation of lay summaries of results, information leaflets and updated safety communications. |
| | Post-study communication: contribution to publications, dissemination of research results to patient community/healthcare professionals. |

**Table 2** Levels of involvement (adapted from the Health Canada Policy Toolkit for Public Involvement in Decision Making (2000))[37]

| Levels of involvement | |
| --- | --- |
| 0 | Did not participate and/or receive any information. |
| 1 | Received information. |
| 2 | Participated in the collection of information. |
| 3 | Participated in the discussion of key points. |
| 4 | Participated in the discussion and actively decided together with other partners. |
| 5 | Participated as a full member of the research team, with equal decision-making power. |

included 42 cancer-related patient organisations, representing individuals such as cancer patients, caregivers and other stakeholders who actively advocate for patients' needs.

## Questionnaire validation

Extensive literature review and consideration of theoretical frameworks were undertaken to identify key dimensions associated with patient's involvement in research. The questionnaire items were thoroughly selected to align with these dimensions, ensuring a comprehensive coverage of the intended construct. Emphasis was placed on clarity, relevance and representativeness of the items to guarantee the questionnaire's effectiveness.

To test the questionnaire's face validity and comprehensiveness, a focus group of seven patient organisation's representatives was organised. Given the small size of the target sample, this focus group was constituted by non-cancer patient organisation's representatives to ensure a similar context. A 3-hour session took place online, providing room for discussion and constructive feedback. The final version of the questionnaire was finalised based on the feedback collected, which proved to be valuable for the instrument improvement.

## Questionnaire distribution

On 10 May 2020, the online questionnaire designed with Qualtrics software was distributed to the 42 cancer-related patient organisations' representatives via email. The objectives of the study and target audience were presented in the introduction followed by a consent request. The questionnaire was open for 2 weeks, and all the organisations were contacted by phone as a follow-up to the initial email. After the first week, all non-responders received a reminder followed by a second phone call and, finally, 2 days before the end date, another email was sent. Participants were not financially compensated, but the working group made a commitment to share and discuss the findings publicly with all participants who showed interest.

## Data handling and reporting

The respondents' data were clustered for analysis, and the identity of each organisation was encrypted and not accessed during data analysis. Only the Principal Investigator had access to the encryption key (automatically generated by the Qualtrics software), with the sole purpose of monitoring the responses and to confirm if the organisations reached were indeed the ones on the target sample. All statistical analyses were performed with IBM SPSS Statistics (V.29.0) and decision-making regarding duplicates and inconsistencies in data provided by the respondents were discussed and unanimously decided in the multistakeholder working group.

## Patient and public involvement

As previously mentioned, this study was ideated, designed and conducted with and for patients since the very beginning. The multistakeholder research team included two patient advocates, one from *EVITA Cancro Hereditário*, a patient organisation focused on hereditary cancer, and another from *Mais Participação, Melhor Saúde*, a community-based action-research collaborative platform.

In 23 June 2020, an online forum driven by some members of the multistakeholder collaborative group and other patient organisations was organised—*New Partners, Better Outcomes—Excellence in Research Through Patient Engagement*—to promote a joint and constructive reflection on the different opportunities for the involvement of patients in research, as well to showcase and share experience-based insights on how partnerships between patients and researchers can help ask better questions, improve research designs, and increase the translation of research findings into the clinic. This forum brought together patient representatives in Portugal, among other healthcare stakeholders, as well as international patient advocates as Jan Geissler (Workgroup of European Cancer Patient Advocacy Networks) and Bettina Ryll (Melanoma Patient Network Europe), and was attended by participants and officials from Germany, Portugal and Slovenia, the countries that composed the Trio Presidency of the Council of the European Union at the time. Key highlights of the study results were briefly discussed to collect contributions for this paper.

## Data analysis

Data were analysed using IBM SPSS Statistics (V.29.0). Statistical analysis included: (1) descriptive statistics (absolute and relative frequencies); (2) $\chi^2$ test of independence for discrete variables or Fisher's exact test of independence, in case of expected frequencies less than 5 in 2×2 contingency tables, to investigate the potential correlation between patient organisation characteristics and their involvement in clinical research; and (3). Cronbach's alpha coefficient and average interitem correlation, to measure the internal consistency (reliability) of the set of survey items related with the organisations' interest to participate and their perceived influence in relation to the 16 R&D opportunities considered. A significance level of 0.05 and 0.001 was used for all statistical tests.

**Table 3** Summary of the characteristics of the 27 Portuguese cancer-related patient organisations that responded to the questionnaire

| Characteristic | | N (%) |
| --- | --- | --- |
| *Scientific advisory board* | | |
| Yes | | 10 (37.0) |
| *Direction board* | | |
| With patient members | | 21 (77.8) |
| *Associate members* | | |
| < 100 members | | 11 (40.7) |
| 100–499 members | | 5 (18.5) |
| 500–2999 members | | 8 (29.6) |
| ≥3000 members | | 3 (11.1) |
| *Annual budget* | | |
| < €100 thousand | | 21 (77.8) |
| €100–250 thousand | | 3 (11.1) |
| €250–500 thousand | | 1 (3.7) |
| €500–1 million | | 0 (0.0) |
| > €1 million | | 2 (7.4) |
| *Regular collaborators* | *Volunteers* | *Employees* |
| 0 | 1 (3.7) | 16 (59.2) |
| 1–3 | 7 (25.9) | 6 (22.2) |
| 4–10 | 11 (40.7) | 2 (7.4) |
| 11–30 | 3 (11.1) | 2 (7.4) |
| > 30 | 5 (18.5) | 1 (3.7) |
| *Collaborative networks* | | |
| National | | 17 (70.0) |
| European/International | | 16 (64.0) |

## RESULTS

### Descriptive analysis of the respondent organisations

Of the 42 cancer-related patient organisations representatives that were contacted, responses were received from 27, meaning a response rate of 64%. A brief summary of the characteristics of these organisations is presented in table 3.

More than half of the respondents identified themselves as the regular representative of the organisation (n=17, 63%). While 10 (37%) have a Scientific Advisory Board, 21 (78%) include patients on their Direction Board and almost half (n=12, 44%) are focused on a single cancer and 5 (19%) on a rare cancer.

There is a great heterogeneity in organisations' lifespan, the oldest having been found in 1941 and the most recent in 2020. The vast majority have an annual budget of up to €100 000 (n=21, 78%) and 2 organisations mentioned having a budget of €1 000 000 or more (n=2, 7%). The three most reported main sources of financing were membership fees (n=22, 85%), individual donations (n=16, 62%) and personal income tax assignment (n=12, 46%).

Regarding associate members, 11 (41%) have less than 100 members. Patients represent more than 60% of the organisation's members for 15 (58%) of the respondents, and healthcare professionals represent 0%–20% of the organisation's members for 25 (96%) of the respondents. The majority (n=16, 59%) does not have any paid workers, operating exclusively based on voluntary work.

Most of the organisations use email (n=25, 96%) and telephone (n=23, 88%) to communicate with their members and 24 (92%) use their own website and Facebook to reach the general community. Twitter is not used by 21 (81%) of the organisations.

When asked about the organisation's main activities, educational events for patients and other audiences (n=21, 78%), support groups (n=18, 67%) and educational materials development (n=17, 63%) were the most reported. Of the 27 organisations, 13 (48%) mentioned having participated in scientific conferences and 7 (26%) in research publications, while 6 (22%) collaborated in research activities. Regarding services provided to members and the cancer community, information about the disease (n=23, 88%), information about patient's rights (n=23, 88%) and psychological support (n=20, 77%) were the most frequently mentioned.

Representatives were also inquired about their preferred sources for building knowledge and how often they access them. The great majority referred using 'Always' or 'Very frequently' the following sources: healthcare professionals (n=26, 96%), websites of international organisations, and conferences and workshops (n=21, 78%), free Internet search (n=20, 74%), websites of national government institutions (n=19, 70%) and scientific publications (n=19, 70%). Scientists and researchers were 'Rarely' or 'Never' a source for building knowledge for 5 (19%) respondents.

Regarding networks and relevant collaborations, 17 (70%) of the organisations integrate a national network and 16 (64%) a European/International network. Collaboration with other patient organisations (n=18, 69%) and the private sector (companies including pharmaceutical) (n=12, 46%) are the most relevant on the day-to-day work of the organisations. Five organisations (19%) reported collaborations with academia and search centres/educational institutions as relevant for their daily work.

### Experience in cancer research

Most organisations (n=20, 74%) have never participated in a clinical study, the most common reason being that they were never invited to do so (n=14, 70%). Of these 20 organisations without previous experience with clinical studies, 16 (80%) also did not participate in other types of health research, such as epidemiological and public health studies.

Of the 7 (26%) organisations having already participated in clinical research, 5 reported having taken the initiative to get involved and 4 of those reported having had the initiative to develop the study itself. On the other

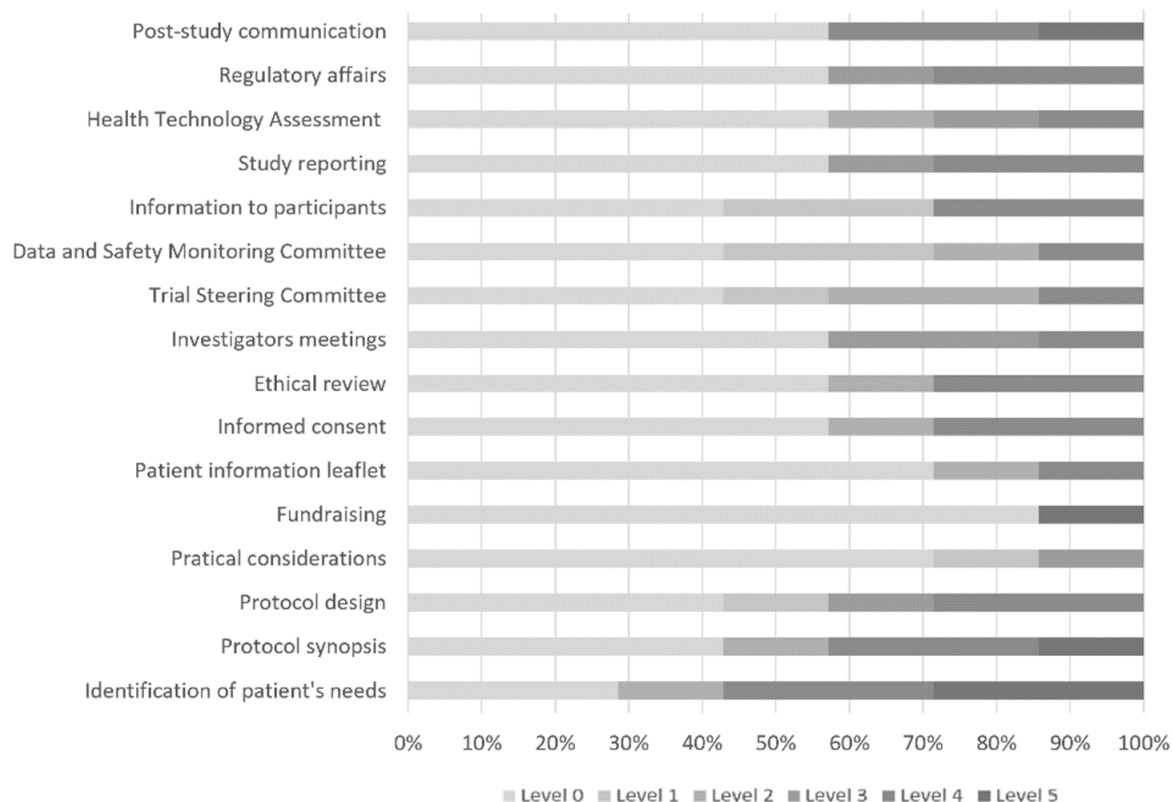

**Figure 2** Respondents' higher level of involvement in the 16 opportunities for patient involvement during the R&D life cycle proposed by Geissler et al.[14]

side, 6 have never participated in the development of clinical recommendations/guidelines.

When enquired about the type of clinical research they were involved in, 6 mentioned having participated in observational studies, 3 in basic or preclinical studies and 2 in clinical trials.

Two organisational characteristics were significantly correlated with patient organisation involvement in clinical research: number of employees (p=0.005) and membership of a European/International network (p=0.035). No correlation (p>0.05) was found for other organisational characteristics (having a scientific advisory board, presence of patients in the direction board, number of associate members, annual budget and membership of a national network).

### Impact and outcomes of research involvement

The seven organisations with research experience were also asked about the higher level of involvement they have ever had in the different stages of the R&D life cycle as proposed by Geissler et al.[14] As shown on figure 2, most of the respondents reported low levels of involvement regarding the 16 different opportunities for patient involvement through the R&D life cycle. More than half of the organisations, 'did not participate or did not receive information' (*Level 0*) in 10 out of the 16 opportunities for patient involvement. However, some participants reported to have been experienced at *Level 5*

('Participated as a full member of the research team, with equal decision-making power'), in activities such as: identification of the real needs of patients (n=2, 29%), study's synopsis design, fundraising for the research project and post-study communication activities (each of the three by n=1, 14%).

In regard to the agreement with different statements concerning the overall participation of the organisation in the clinical studies, most respondents mention having a positive perception of that involvement. More than half of the organisations 'Agrees' or 'Strongly agrees' that: their participation has been valued by the research team during the study (n=5, 71%); their contribution has been integrated (n=5, 71%); was recognised as a partner of equal relevance in the study (n=5, 71%) and has actively participated in the research (n=4, 57%). On the other side, 3 (43%) organisations 'Disagrees' or 'Strongly disagrees' that, after being involved in the study, the results were shared with the organisation, before being published.

Regarding the overall impact of their involvement, most respondents identified that their participation contributed to increase the patient recruitment rate (n=5, 71%), to help the patients involved with a better understanding of the study informed consent (n=4, 57%), to increase the quality of the generated knowledge and to decrease the time needed to complete the study (both n=3, 43%).

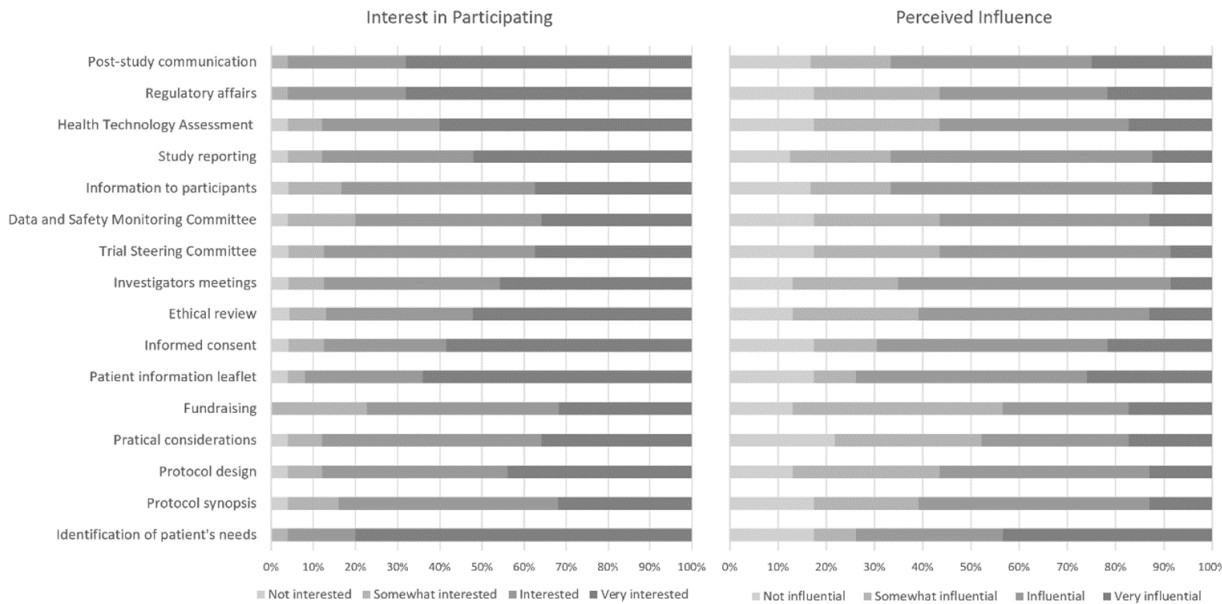

**Figure 3** Respondents' interest in participating and perceived level of influence in the 16 opportunities for patient involvement during the R&D life cycle proposed by Geissler *et al*.[14]

## Views and motivations on patient involvement in cancer research

Regardless of whether, or not, organisations have had previous experience in participating in clinical research, all respondents were asked about the level of interest of their organisations in participating in each step of the R&D life cycle and the level of influence they perceived to have in each step. Results are in figure 3. More than half of the organisations is 'Interested' or 'Very interested' in being involved in any of the different R&D life cycle steps. For the set of 16 steps, Cronbach's alpha was 0.929 and average interitem correlation was 0.461, indicating an acceptable internal consistency. In particular, more than half is 'Very interested' in being involved in the following phases: identification of the real needs of patients (n=20, 74%), regulatory affairs and post-study communication activities (both n=17, 63%), patient information sheet about the project (n=16, 59%), Health Technology Assessment (n=15, 56%) and informed consent (n=14, 52%). However, for the same opportunities for patient involvement in the R&D life cycle, lower levels of perceived influence were identified and Cronbach's alpha was 0.987 and average interitem correlation was 0.830, suggesting that the items are highly intercorrelated and hence redundant.

When respondents were asked about their motivations to be involved in clinical studies, regardless of whether they have had previous experiences, or not, 20 (74%) reported wanting to ensure that the study and its results address true patients' needs, 15 (56%) to develop healthcare and therapies that are more representative of patients' true needs and 13 (48%) to apply patients' experience and knowledge about their condition.

## DISCUSSION

Only 26% of the respondent's Portuguese cancer-related patient organisations were ever involved in a clinical study at the time we conducted our survey. This is in line with previous findings from the literature,[14–17] that despite an increased interest and research on involvement frameworks and benefits, and the issuing of practical recommendations by various stakeholders, patient involvement in research is not yet a standard, meaningful and systematic practice.

In addition, although organisations with prior research experience have a positive perception on the way they were involved and how their participation contributed to more favourable research outcomes, a low level of involvement in the various R&D steps was actually observed. This finding is corroborated by the low level of perceived influence reported by the organisations participating in the study, despite the high reported motivation to participate in the different R&D steps and the perceived benefits of the organisation's involvement in research. The findings suggest that meaningful involvement in research has not yet been fully appropriated by patient organisations, neither by researchers and other stakeholders involved, and consequently not completely put into practice. The gap analysis in patient involvement practices conducted by Faulkner *et al*[29] lead to clear directional insights to enhance collaborative practices that might mitigate this, such as empowerment of stakeholders (both patients and researchers) focusing on training on their roles and responsibilities. Furthermore, awareness raising for principles that should guide patient involvement and the benefits of patient-centred research may contribute to increasing the actual level of patient involvement in the various R&D steps.

Most of the organisations, when asked about its main activities and services, identified patient's support and educational related ones as the most representative, which goes in line with the conventional understanding of a patient organisation's mission and the findings of Amaro *et al* in 2015.[33] However, it is relevant to outline from our results that five of the seven organisations involved in research reported having taken the initiative to get involved, and four of those reported having had the initiative to develop the study itself. This suggests that these organisations either are not involved in research at all or, if they do, they are the ones who promote it. Our findings do not allow us to understand *how* in fact these organisations take the initiative to develop, or to be involved, in clinical studies and it would be very relevant, in future work, to understand how these partnerships between patients and researchers were initiated to provide the literature with examples of best practices and recommendations.

Moreover, when asked about *why* some organisations had never participated in a clinical study before, the most outlined reason was the fact that the organisation was never invited to participate in this type of study, and this may represent a critical reason for the low involvement of these organisations in clinical studies: researchers are not reaching them. Our findings do not allow us to understand *why* and *what* is missing in the communication between patients and researchers, and therefore it would be relevant, as future work, to understand how these organisations could make their interest in being partners in research projects more visible and clearer, and what is preventing them from having a more active role in taking the initiative to be involved in research. The lack of fully dedicated and professionalised teams, as our results suggest, may play a relevant role in this. Also, developing a questionnaire with mirrored questions to this one, addressed to researchers would be essential to understand the researchers' side and perspective on this matter.

Regardless of having had previous experience with clinical studies, our results also reveal that most Portuguese cancer-related patient organisations have significant interest when asked about being involved in different stages of a clinical study. Ensuring that research results are more aligned with the real needs of patients are the main reported motivations. These findings are particularly relevant for researchers to be aware that these organisations are willing to collaborate in research. The motivation expressed by all organisations to participate in clinical research can and should be turned into greater and more meaningful involvement in practice, so that the cancer community can benefit from the outcomes of a truly patient-centred research.

When asked about the global impact of their participation in the studies, more than half of the seven organisations that were, or have ever been, involved in clinical studies claimed to have contributed to the increase in the participation rate of patients and a better understanding of informed consent by the patients involved. This is an important finding for the many researchers that often struggle with low retention rates in clinical studies and with the designing of an informed consent in lay language that is clear and enlightening for the patients.

Furthermore, it might be worth highlighting that the results obtained from this survey contrast significantly with the results obtained by Landy *et al* in 2013.[38] The latter targeted international genetic diseases advocacy organisations to understand how they participate in clinical research and found that the majority of these organisations participate directly in multiple aspects of research, ranging from study design and patient recruitment to data collection and analysis. However, our results are more aligned with the findings of Halvorsrud *et al*[26] and Pii *et al*,[27] where evidence showed that cocreation was rarely extended to later stages of research and public and patient involvement in cancer research has especially integrated in the early stages of the research process, in defining and prioritising research. This disparity in patient involvement through the R&D life cycle might be due to several reasons, and, in the future, it could be relevant to analyse different factors that might have an impact on it, considering different countries and disease areas, such as the type and prevalence of a disease. Our results suggest that participation in European/International research networks is a factor that may play a role in boosting patient involvement in cancer research.

Unlike earlier studies, a key aspect of our research is the in-depth examination of patient involvement across different stages of the R&D process. While prior studies focused on identifying whether patients are part of different R&D steps, our study also analysed the extent of their participation. This methodological refinement contributes to a more profound comprehension of how patients participated throughout the R&D continuum.

Regarding the organisations' characterisation, this study provides a very complete overview of the Portuguese cancer-related patient organisations, never conducted before, on important subjects such as: organisational structure, financial sources, members profile, activities and services, communication channels and collaborations. However, due to the small universe of Portuguese cancer-related patient organisations and the low number of organisations reporting previous involvement in clinical research, we acknowledge that the conducted descriptive analysis provides little information on the correlation of the organisations' characteristics and their involvement in clinical research. Our analysis highlights that the number of employees (paid collaborators) and integration in a European/International network are factors that may play a role on the higher involvement of patients in research. The first factor may contribute to the necessary infrastructure in support of involvement in research activities, and the latter may enable the organisation to become more visible to potential partners and more aware of opportunities for involvement. In future research, it is worth further investigating the role that distancing

between patient organisations and academia (universities, through contact with scientists and researchers, were 'rarely' or 'never' a preferred source for building knowledge by 19% of the organisations) may play.

### Study limitations

While achieving a 64% response rate for the questionnaire, this pilot study's scope was confined to Portuguese cancer-related patient organisations, a small universe. Given the low number of organisations with prior involvement in clinical research, the descriptive analysis provides limited insights into the correlation between organisational characteristics and participation in clinical research. To be able to draw broader conclusions, this questionnaire should be applied to a broader scope, allowing to explore these correlations along with variations across countries and disease areas.

Our findings do not offer an understanding of the reasons behind communication gaps between patients and researchers, also it falls short in elucidating how these patient organisations initiate or become involved in clinical studies. Future research should focus on unravelling the mechanisms behind the initiation of partnerships between patients and researchers.

Acknowledging the absence of the researchers' perspective in this study, we emphasise the importance of including it, as future work, for a more thorough understanding of patient involvement in cancer research. This can be achieved by developing a mirrored questionnaire specifically designed for researchers.

## CONCLUSIONS

This study provides the first in-depth characterisation of Portuguese cancer-related patient organisations and their views, motivations and experiences on patient involvement in cancer research. Most importantly, this study revealed that most Portuguese cancer-related patient organisations were never involved in research but show high interest to be involved in different stages of a clinical study to ensure that research results are more aligned with the real needs of patients. The motivation expressed by all organisations to participate in clinical research should be turned into greater and more meaningful involvement in practice, through the develop of advocacy campaigns and public policies on patient involvement in cancer-research, so that the cancer community can benefit from the outcomes of a truly patient-centred research.

In the future, this questionnaire can be applied to other disease and umbrella patient organisations, at national and/or international levels, since it is not specific to any disease or the local context, contributing to a broader understanding of their and their views, motivations and experiences on patient involvement in research.

**Contributors** CR, TM and AV conceptualised the study. All authors contributed to study design. CR, TM, RSR, ARP and AV contributed to data collection and CR, SC, RSR and ARP contributed to data analysis. CR and SC contributed to manuscript preparation. CR, SC, TM and RSR reviewed, edited and approved the final manuscript. ARP and AV reviewed and approved the final manuscript. CR is responsible for the overall content as the guarantor.

**Funding** This work was funded by Fundação para a Ciência e a Tecnologia (UIDB/00124/2020, UIDP/00124/2020 and Social Sciences DataLab - PINFRA/22209/2016), POR Lisboa and POR Norte (Social Sciences DataLab, PINFRA/22209/2016).

**Competing interests** None declared.

**Patient and public involvement** Patients and/or the public were involved in the design, or conduct, or reporting, or dissemination plans of this research. Refer to the Methods section for further details.

**Patient consent for publication** Not applicable.

**Ethics approval** This study, and the questionnaire, received the approval of the Ethics Committee of the NOVA School of Business and Economics on 7 April 2020. Participants gave informed consent to participate in the study before taking part.

**Provenance and peer review** Not commissioned; externally peer reviewed.

**Data availability statement** All data relevant to the study are included in the article or uploaded as supplementary information.

**ORCID iDs**
Constança Roquette http://orcid.org/0009-0004-5964-6268
Sofia Crisóstomo http://orcid.org/0000-0002-1907-1883
Ana Rita Pedro http://orcid.org/0000-0002-9197-7129

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
