## [Reviewer comments · BMJ Open]

ARTICLE DETAILS

TITLE (PROVISIONAL)	Patient Organizations' Views, Motivations and Experiences on Patient Involvement in Cancer Research - A Pilot Study in Portugal
AUTHORS	Roquette, Constança; Crisóstomo, Sofia; Milagre, Tamara; Ribeiro, Rute; Pedro, Ana Rita; Valente, André

VERSION 1 – REVIEW

REVIEWER	Zrínyi, Miklós University of Pécs
REVIEW RETURNED	26-Jul-2023

GENERAL COMMENTS	The topic explored in your paper is important and deserves to be published. Please consider making suggested revisions as per below. 1) Please revise your introduction to account for the larger number of studies reported in the literature and draw conclusions that may be relevant to explain the need and direction for your research. I ran a simple Pubmed search and found these examples: Schuster ALR, Hampel H, Paskett ED, Bridges JFP. Rethinking Patient Engagement in Cancer Research. Patient. 2023 Mar;16(2):89-93. doi: 10.1007/s40271-022-00604-9. Epub 2022 Oct 27. PMID: 36301439; PMCID: PMC9911482. A Spears P. Patient engagement in cancer research from the patient's perspective. Future Oncol. 2021 Oct;17(28):3717-3728. doi: 10.2217/fon-2020-1198. Epub 2021 Jul 2. PMID: 34213358. Schandl A, Mälberg K, Haglund L, Arnberg L, Lagergren P. Patient and public involvement in oesophageal cancer survivorship research. Acta Oncol. 2022 Mar;61(3):371-377. doi: 10.1080/0284186X.2021.2016950. Epub 2021 Dec 20. PMID: 34923913. 2) Please address in more detail your efforts to establish as much validity as possible for your self-developed main measure. Also, do report your reliability coefficient (Cronbach's alfa) in relation to how well your instrument performed. 3) Please include a description of your statistical analyses under the method section. If appropriate, I would welcome more complex analyses (uni- or multivariate) beyond descriptive outcomes.
--

	4) Please include a 'limitations' section at the end of your paper to acknowledge the potential shortcomings of your research (i.e., sample size, instrument validity etc.). 5) Please compare and contrast your research findings to earlier studies found in the literature (see my 1) comment) and revise your discussion accordingly. Thank you for considering the above and good luck.
--	---

REVIEWER	Mateus, Ceu Lancaster University, Division of Health Research
REVIEW RETURNED	18-Oct-2023

GENERAL COMMENTS	Thank you very much for the opportunity to review this paper. p2 line 39 - I think 'disease' should be plural Methods: it would be helpful to have more details on 'the multi-stakeholder collaborative group was put in place with both patient representatives and researchers'. Like size, how many participants from patient representatives and researchers, affiliations of researchers, and so on. p5 - line 34: consider using 'agreed' instead of 'determined' Ethics approval - was the study also approved by the ethics committee of NOVASBE or only the questionnaire? p8 - the first sentence is difficult to understand: 'The respondents' data were clustered for analysis, without assessing each organization's performance being assessed separately' p9 - how many participants were in the multi-stakeholder team? How many participants from each organisation? this statement is not part of the methods: 'This partnership with patient representatives and researchers contributed to a more accurate measurement instrument and more impactful study outcomes.' This is a result. Can your methods confirm this result? I suggest including a table with the descriptive results of the respondents in the results section. p9 - can you please indicate other types of health research participants were asked about? 'did not participate in other types of health research' such as? In the discussion, it is never mentioned if patients' associations approach researchers to let them know they exist and are willing to take part in research projects and are never contacted back. Do the associations make themselves known? Do they make it clear they are interested and have the infrastructure to be partners in research projects? or do they take a passive role and expect to be 'found' by researchers? please consider adding the questionnaire as an online resource for the readers.
---

VERSION 1 – AUTHOR RESPONSE

viewer: 1 Dr. Miklós Zrínyi, University of Pécs	1 Please revise your introduction to account for the larger number of studies reported in the literature and draw conclusions that may be relevant to explain the need and direction for your research. I ran a simple Pubmed search and found these examples: Schuster ALR, Hampel H, Paskett ED, Bridges JFP. Rethinking Patient Engagement in Cancer Research. Patient. 2023 Mar;16(2):89-93. doi: 10.1007/s40271-022-00604-9. Epub 2022 Oct 27. PMID: 36301439; PMCID: PMC9911482. A Spears P. Patient engagement in cancer research from the patient's perspective. Future Oncol. 2021 Oct;17(28):3717-3728. doi: 10.2217/fo-2020-1198. Epub 2021 Jul 2. PMID: 34213358. Schandl A, Mälberg K, Haglund L, Arnberg L, Lagergren P. Patient and public involvement in oesophageal cancer survivorship research. Acta Oncol. 2022 Mar;61(3):371-377. doi: 10.1080/0284186X.2021.2016950. Epub 2021 Dec 20. PMID: 34923913.	Additional 14 relevant references were added to this manuscript in order to provide an updated overview of the cancer research landscape and to better contextualize our research.
	2 Please address in more detail your efforts to establish as much validity as possible for your self-developed main measure. Also, do report your reliability coefficient (Cronbach's alfa) in relation to how well your instrument performed.	The Questionnaire validation section was updated, and Data Analysis section was added (prior to Results). In Results section, we added some Chi-Square and Reliability tests.
	3 Please include a description of your statistical analyses under the method section. If appropriate, I would welcome more complex analyses (uni- or multivariate) beyond descriptive outcomes.	
	4 Please include a 'limitations' section at the end of your paper to acknowledge the potential shortcomings of your research (i.e., sample size, instrument validity etc.).	Limitations section was added following the Discussion section. Please see the new version of the manuscript.

	5 Please compare and contrast your research findings to earlier studies found in the literature (see my 1) comment) and revise your discussion accordingly.	The Discussion was fully revised in order to include a comparison and contrast to other relevant and up to date research findings. Please see the new version of the manuscript.
Reviewer: 2 Prof. Ceu Mateus, Lancaster University	1 p2 line 39 - I think 'disease' should be plural	The Abstract was fully revised, and the sentence where this typo was is not included anymore. Please see the new version of the manuscript.
	2 Methods: it would be helpful to have more details on 'the multi-stakeholder collaborative group was put in place with both patient representatives and researchers'. Like size, how many participants from patient representatives and researchers, affiliations of researchers, and so on.	The following alterations were made: "In December 2020, encouraged by the Europe: Unite against Cancer declaration, a multi-stakeholder collaborative group of 6 people was put in place with both patient representatives and researchers (...). The multi-stakeholder research team included two patient advocates, one from EVITA Cancro Hereditário (a patient organization focused on hereditary cancer) and another from Mais Participação, Melhor Saúde (a community-based action-research collaborative platform), as well as three researchers from Universidade NOVA de Lisboa (NOVA School of Business and Economics and the Portuguese National School of Public Health) and one researcher from the Champalimad Foundation. "
	3 p5 - line 34: consider using 'agreed' instead of 'determined'	The sentence was changed to "After several work sessions, the questionnaire structure was agreed "
	4 Ethics approval - was the study also approved by the ethics committee of NOVASBE or only the questionnaire?	The study was approved along with the questionnaire. We have made the following changes on the manuscript to clarify it: "This study, including the questionnaire, received the approval of the Ethics Committee of the NOVA School of Business and Economics on April 7th, 2020."
	5 p8 - the first sentence is difficult to understand: 'The respondents' data were clustered for analysis, without assessing each organization's performance being assessed separately'	The sentence was changed to "The respondents' data were clustered for analysis, and the identity of each organization was encrypted and not accessed during data analysis. "

6	p9 - how many participants were in the multi-stakeholder team? How many participants from each organisation?	As mentioned above we have changed the Methods section to provide this information: The following alterations were made: "In December 2020, encouraged by the Europe: Unite against Cancer declaration, a multi-stakeholder collaborative group of 6 people was put in place with both patient representatives and researchers (...). The multi-stakeholder research team included two patient advocates, one from EVITA Cancro Hereditário (a patient organization focused on hereditary cancer) and another from Mais Participação, Melhor Saúde (a community-based action-research collaborative platform), as well as three researchers from Universidade NOVA de Lisboa (NOVA School of Business and Economics and the Portuguese National School of Public Health) and one researcher from the Champalimaud Foundation. "
7	this statement is not part of the methods: 'This partnership with patient representatives and researchers contributed to a more accurate measurement instrument and more impactful study outcomes.' This is a result. Can your methods confirm this result?	This sentence was removed from the Methods section.
8	I suggest including a table with the descriptive results of the respondents in the results section.	With added Table 3 with a summary of the main characteristics of respondent organizations.
9	p9 - can you please indicate other types of health research participants were asked about? 'did not participate in other types of health research' such as?	Participants were asked "Has your organization ever participated in another type of health research? (epidemiological studies, public health, etc.)". We have made the following alterations: "Of these 20 organizations without previous experience with clinical studies, 16 (80%) also did not participate in other types of health research, such as epidemiological and public health studies."

	In the discussion, it is never mentioned if patients' associations approach researchers to let them know they exist and are willing to take part in research projects and are never contacted back. 10 Do the associations make themselves known? Do they make it clear they are interested and have the infrastructure to be partners in research projects? or do they take a passive role and expect to be 'found' by researchers?	We have an interesting result regarding this: Of the 7 (26%) organizations having already participated in clinical research, 5 (71%) reported having taken the initiative to get involved and 4 (80%) of those reported having had the initiative to develop the study itself. Changes were made in the Discussion to make these questions addressed in a clearer way. Please see the new version of the manuscript.
	11 please consider adding the questionnaire as an online resource for the readers.	Please find it now attached as supplementary material.

VERSION 2 – REVIEW

REVIEWER	Zrínyi, Miklós University of Pécs
REVIEW RETURNED	09-Dec-2023

GENERAL COMMENTS	Thank you for the revisions, the manuscript has improved significantly and now has more clarity.
--